# The structure of a plant-specific partitivirus capsid reveals a unique coat protein domain architecture with an intrinsically disordered protrusion

Matthew Byrne [1✉], Aseem Kashyap[2], Lygie Esquirol[2], Neil Ranson [1] & Frank Sainsbury [2,3✉]

Persistent plant viruses may be the most common viruses in wild plants. A growing body of evidence for mutualism between such viruses and their hosts, suggests that they play an important role in ecology and agriculture. Here we present the capsid structure of a plant-specific partitivirus, Pepper cryptic virus 1, at 2.9 Å resolution by Cryo-EM. Structural features, including the $T = 1$ arrangement of 60 coat protein dimers, are shared with fungal partitiviruses and the picobirnavirus lineage of dsRNA viruses. However, the topology of the capsid is markedly different with protrusions emanating from, and partly comprising, the binding interface of coat protein dimers. We show that a disordered region at the apex of the protrusion is not required for capsid assembly and represents a hypervariable site unique to, and characteristic of, the plant-specific partitiviruses. These results suggest a structural basis for the acquisition of additional functions by partitivirus coat proteins that enables mutualistic relationships with diverse plant hosts.

[1] Astbury Centre for Structural Molecular Biology, Faculty of Biological Sciences, University of Leeds, Leeds, UK. [2] Centre for Cell Factories and Biopolymers, Griffith Institute for Drug Discovery, Griffith University, Nathan, QLD 4111, Australia. [3] Synthetic Biology Future Science Platform, Commonwealth Scientific and Industrial Research Organization (CSIRO), Brisbane, QLD 4001, Australia. ✉email: M.J.Byrne@leeds.ac.uk; F.Sainsbury@griffith.edu.au

Despite the premise that persistent viruses of plants can be beneficial to their hosts, they are poorly understood[1]. Previously known as cryptic viruses, they have until recently not been known to induce observable symptoms in infected plants. Nevertheless, they infect many horticultural crops and metagenomic studies suggest that persistent viruses, such as members of the *Partitiviridae*, are amongst the most common viruses in wild plants[2]. Mutualistic or symbiotic viruses have been described for all domains of life[3] and a general recognition that beneficial viruses provide selective advantage to their hosts is emerging. Therefore, the study of persistent plant viruses is important in understanding the ecological niche of their wild plant hosts, as well as in supporting the development and management of agricultural varieties.

The *Partitiviridae* are a family of viruses that persistently infect fungi, protozoa, plants and arthropods. Viruses in this family are encoded by small, bi-segmented or tri-segmented, double-stranded RNA (dsRNA) genomes, with each segment coding for a single gene, an RNA-dependent RNA polymerase (RdRp) and one or two putative coat proteins (CPs). Transmission of partitiviruses occurs strictly vertically via meiosis; there does not appear to be systemic movement and there are no known natural vectors[4]. The family is currently divided across five genera, grouped according to the similarity of their RdRp genes[5,6]. Members of the *Alphapartitivirus* and *Betapartitivirus* infect fungi or plants, the *Gammapartitivirus* infect fungi, the *Deltapartitivirus* infect plants and the *Cryspovirus* infect protozoa. Two high-resolution gammapartitivirus structures[7,8] and a low-resolution betapartitivirus structure[9] show that the partitiviruses share the typical capsid arrangement common to the dsRNA viral lineage, with 60 CP homodimers arranged with $T = 1$ icosahedral symmetry.

The question of how partitiviruses develop mutualistic relationships with their plant hosts is central to understanding the impact they have on both domesticated and wild plants species. In at least one case, a beneficial effect on the host plant has been ascribed to the CP of a plant-associated partitivirus; recombinant expression of the CP from white clover cryptic virus, an alphapartitivirus, suppresses nodulation in legumes when adequate nitrogen is present in the soil[10]. The acquisition of additional functions is common to CPs of plant viruses[11] and the lack of extracellular phase or cell-to-cell movement by the partitiviruses may facilitate the acquisition of specific host interactions not directly involved in uptake, replication or movement. On the other hand, the partitiviruses must maintain a functional and intact capsid to prevent host recognition of the dsRNA genome and co-localise the RdRp in a protective compartment for genome replication. Most fungal dsRNA viruses also lack an extracellular phase, and a mechanism for the insertion of acquired function in permissive sites around a conserved α-helical CP domain has been proposed[12,13]. The most well-characterised example, is the enzymatic mRNA decapping activity of the yeast L-A virus CP[14].

In this study, we utilise transient expression of virus-like particles (VLPs) in plants and high-resolution cryo-EM to determine the structure of Pepper cryptic virus 1 (PCV-1). To our knowledge, there are no plant-specific partitivirus structures publicly available. PCV-1 is ubiquitous in Jalapeno peppers (*Capsicum annum*), and is vertically transmitted with an efficiency of > 98%[4] without causing apparent disease symptoms. However, PCV-1 infection reduces aphid feeding, which may decrease the likelihood of transmission of acute viruses that do cause symptoms and thus impact crop yields[15]. The overall architecture of the isometric capsid is similar to the available structures of fungal partitiviruses and the picobirnavirus lineage of dsRNA viruses[13,16] with 60 coat protein dimers arranged in $T = 1$ symmetry. However, the surface topology is strikingly different with capsid protrusions emanating from the dimer interface. We identify a disordered region that forms the outermost portion of the deltapartitivirus capsid protrusion and is characteristic of the plant-specific genus. The disordered region is hypervariable and we show that it is not required for assembly of virus-like particles. We hypothesise that this region provides a permissive site for insertions that enable the symbiotic relationship between deltapartitiviruses and diverse dicotyledonous hosts.

## Results and discussion

**Expression and characterisation of deltapartitivirus particles.** The capsids of persistent plant viruses are found at very low levels; therefore, to allow for determination of the PCV-1 capsid structure, we expressed PCV-1 CP in *Nicotiana benthamiana* using the pEAQ-*HT* vector[17]. Virus-like particles (VLPs) were purified from infiltrated leaves with a single ultracentrifugation step, resulting in a single protein of $MW_r$ ~47.5 kDa, which was confirmed as the PCV-1 CP by mass spectrometry and initially analysed by negative stain TEM to ensure particle integrity (Supplementary Fig. 1). PCV-1 VLPs formed homogenous, monodisperse particles of approximately 35 nm in diameter. Transient expression in plants has become a viable alternative for eukaryotic protein production, and is particularly well suited to VLP production[18–20]. Here it resulted in simple recovery of the PCV-1 capsid, a virus known to exist at low titres in natural infections[4]. We have previously shown that VLPs expressed in plants act as good facsimiles, in place of virions, for high-resolution capsid structure determination[21–23]. Furthermore, recombinant expression of VLPs provides a means to isolate mutations to structural proteins from the viability of the virus, enabling the study of mutations that affect genome interactions[21,24] and capsid assembly[23,24].

In an effort to determine the high-resolution structure of the PCV-1 capsid, cryo-EM grids of PCV-1 VLPs suspended in vitreous ice were prepared and visualised (Fig. 1a). Structure refinement with icosahedral symmetry imposed yielded a density map with a global resolution of 2.9 Å (Fig. 1b). Local resolution extends to 2.75 Å in the majority of the shell domain (Supplementary Fig. 2), allowing for accurate de novo modelling of the PCV1 structure. Final models were built for a single asymmetric unit and refined such that the model was in the presence of all surrounding chains to satisfy inter-chain interactions and avoid clashes. The final atomic model of PCV1 comprises residues 29–328 and 379–411. The reconstruction has no density for the intervening region (residues 329–378). This is likely due to a high degree of structural variability, indicating that this portion of the capsid likely forms a flexible loop. Likewise, there is no density for the N-terminus (residues 1–28), indicating that this region is also highly flexible. The structure of PCV-1 reveals a capsid structure composed of 120 protomers arranged with $T = 1$ symmetry, with each asymmetric unit comprising a homodimer of the PCV-1 CP. Superposition of the individual chains comprising a single asymmetric unit yields an RMSD of 0.7 Å (Supplementary Fig. 3). Variations in backbone are most prominent nearer the 5-fold and 3-fold symmetry axes, as would be anticipated for quasi-conformers. There is no density at the interior of the capsid, indicating that PCV1 VLPs do not non-specifically encapsidate endogenous RNAs, as has been observed for other VLPs produced in plants, though not those from other dsRNA viruses[25].

The CP fold is made up of two distinct domains, the shell (S) and protrusion (P) domains. The S domain is made up primarily of α-helices (secondary structure nomenclature detailed in Supplementary Fig. 4 for reference). Seven α-helices surround a central, longer α-helix (α3) such that the S domain is roughly

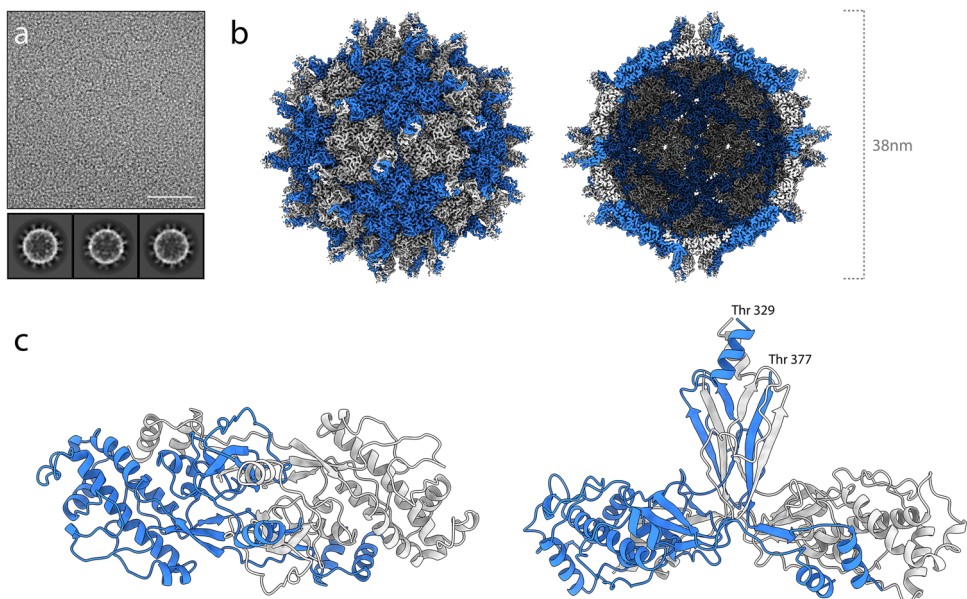

**Fig. 1 The cryo-EM structure of PCV-1. a** A section of a representative micrograph of PCV-1 and 2d class averages. **b** The 2.9 Å resolution 3D reconstruction of PCV-1, coloured according to quasi-conformer, with monomer A coloured blue and monomer B coloured grey, (left) capsid surface (right) central slice through capsid. **c** The atomic model of PCV-1, showing the model for a single asymmetric unit, comprising A/B quasi-conformers coloured blue and grey, respectively.

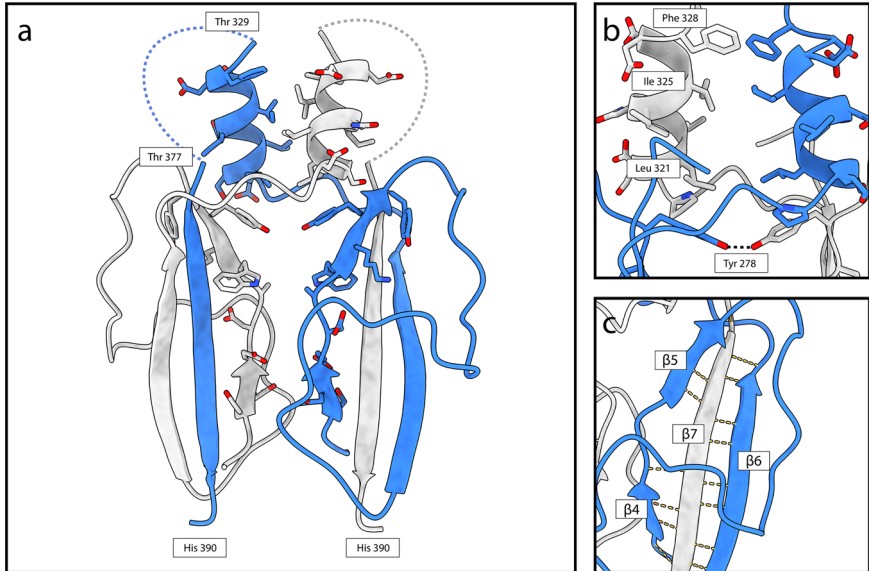

**Fig. 2 The PCV-1 protrusion domain forms the homodimer interface. a** The PCV-1 P domain shown in cartoon form, residues close to the interface are shown in stick form. Dashed lines denote the regions occupied by the hypervariable loop for which there is no density. **b** α7 from each subunit form a hydrophobic interface at the tip of the P domain. **c** β7 from each subunit swaps into the opposing P domain forming an extended hydrogen bond network in a 3-standed β-sheet.

rhomboidal in shape. In PCV-1, α1 (res. 33–41) domain swaps into the S domain of the opposing protomer within the same asymmetric unit via an extended loop (res. 42–56). The protrusion comprises a 3-stranded beta sheet (β4-7), in which the central β-stand (β7) is domain swapped from the opposing monomer, and a α-helix (α7) (Fig. 2a). The P domain (res. 269–389) forms a large section of the homodimer interface, with a buried surface area of 3104 Å². Similarly, strong interactions in other partitiviruses and picobirnaviruses have led to the suggestion that the CP dimer is the basic assembly unit[13,26] and that also appears to be the case with PCV-1. Opposing chains of the protrusion meet at their tip and base, with a small gap

through the centre. At the tip, α7 from opposing monomers wrap around one another, interacting across a hydrophobic interface (Fig. 2b). Extensive hydrogen bonding along the domain swapped β7 forms the majority of the polar interactions at the dimer interface (Fig. 2c). Following the P domain, a short loop (res. 380–389) and an α-helix (α8) domain swap into the opposing monomer's S domain such that the N and C termini of each subunit sit close to one another, as is seen in all other partitivirus structures to date.

**Comparison to other partitiviruses.** Three-dimensional structures have been determined for three Partitiviruses to date, the

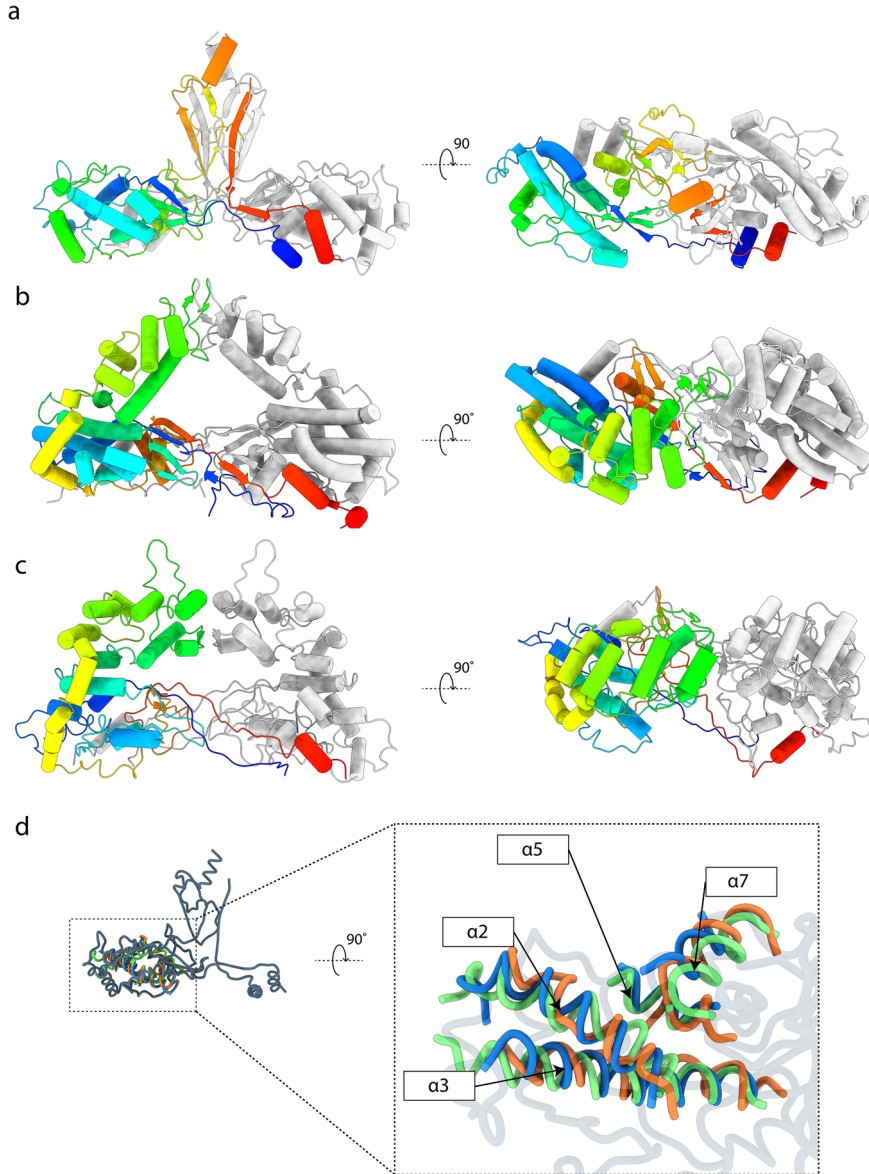

**Fig. 3 Comparison of the PCV1 structure with other partitiviruses.** For each virus a quasi-symmetric homodimer is shown in cartoon representation, with subunit A coloured from N-terminus (blue) to C-terminus (red), and subunit B coloured grey for clarity. **a** The atomic model of PCV-1. **b** The atomic model of PsV-F (PDB 3ES5). **c** The atomic model of PsV-S (PDB 3IYM). **d** PCV-1 shell domain superposed with core α-helices from the PsV-F/PsV-S shell domains. Models shown in ribbon format. PCV-1 is coloured grey. PsV-F is coloured green. PsV-S is coloured orange. Common α-helices in PCV-1 coloured blue for clarity. α2, α3, α5 and α7 from PCV-1 are superposed with α1, α2, α4 and α14 from PsV-F and PsV-S.

Penicillium stoloniferum viruses F and S (PsV-F, PsV-S) from the genus *Gammapartitivirus*[7,8,27], and Fusarium poae virus 1, from the genus *Betapartitivirus*[9]. All three infect fungi, and share common structural features, including 120 copies of a roughly rhomboid shaped CP arranged in a $T = 1$ icosahedron. In all three structures, the P domains protrude distally, meeting at the centre of the asymmetric unit and forming a large portion of the homodimer interface. Only PsV-F and PsV-S have models deposited to the PDB and the comparison of these with PCV-1 is shown in Fig. 3. In all three structures, the shell domain is composed of primarily α-helical bundles, in which one longer α-helix (α3 in PCV-1, but α2 in PsV-F and PsV-S) is surrounded by a number of shorter α-helices. Interestingly, α2, α3, α5 and α7 from the PCV-1 shell domain superpose onto α1, α2, α4 and α14 respectively, of PsV-F and PsV-S, suggesting that these four helices may describe the minimal fold for the prototypical rhomboid shell domain of partitiviruses (Fig. 3d). The other

helices of the PCV-1 and PsV shell domains do not superpose with any discernible similarity. Elements of the minimal prototypical shell domain described here can be seen in other coat proteins of viruses that lie within the previously identified dsRNA CP viral lineage[13,16]. When superposed with the structure of the CP of Picobirnavirus[28], for example (PBD 2VF1), α3, α5 and α7 from PCV-1 overlay with corresponding structural elements, suggesting a common ancestor and demonstrating the utility of the core of the dsRNA shell domain. Strikingly, the protrusion domain of PCV-1 is markedly different from those of PsV-F and PsV-S in its overall architecture. It does not form an arch-like structure as in PsV-F/S, but rather emerges from the homodimer interface to form 60 spike-like features on the capsid surface (Fig. 1). Furthermore, the P domain of PCV-1 is largely β-sheet, where those of PsV-F and PsV-S are largely α-helical (Fig. 3). In PCV-1, following the disordered region of the protrusion (Fig. 2), it is likely that the protein backbone domain swaps back into the

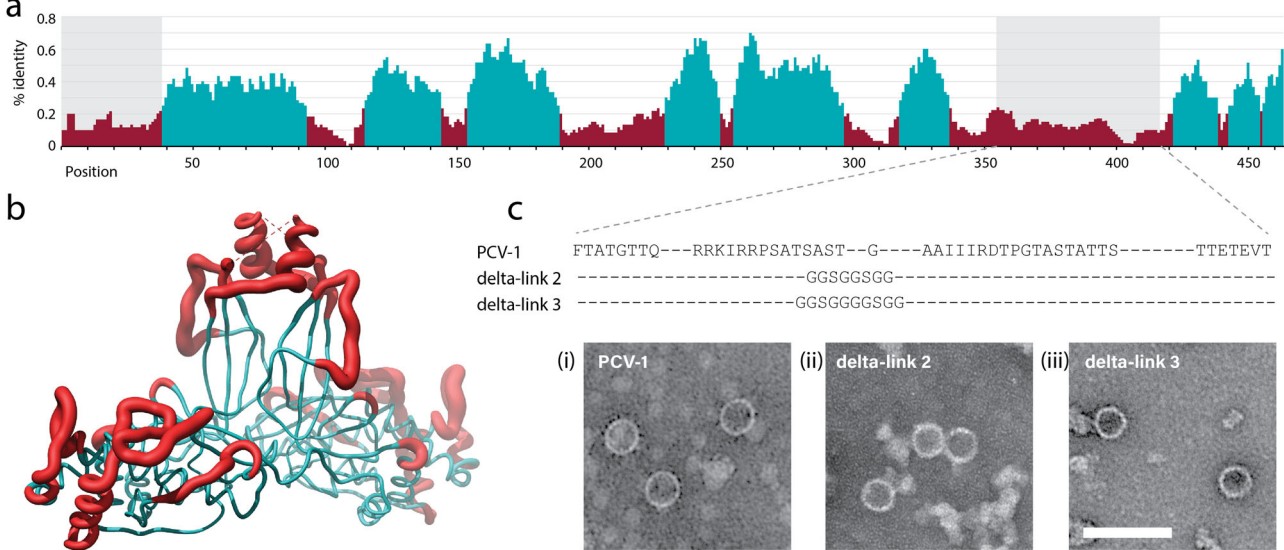

**Fig. 4 The hypervariable protrusion is not required for capsid assembly. a** Alignment of CP amino acid sequences of PCV-1, Pepper cryptic virus 2 (PCV-2) and the two CP sequences identified for Beet cryptic virus 2 (BCV-2). Shaded regions correspond to the unresolved regions of PCV-1 CP and positions coloured cyan correspond to > 25% identity using a 10-position rolling average. **b** Map of the percentage identity from the alignment to the PCV-1 protomer structure. Cyan regions have ≥ 25% identity and red regions have < 25% identity (10-position rolling average) with grey shading indicating the missing density in the PCV-1 reconstruction. **c** PCV-1 CP variants with protrusion deletions and TEM images of density gradient purified particles. Images show (i) WT PCV-1 CP, and two deletions of the disordered region, replaced with short linker sequences delta-link 2 (ii) and delta-link 3 (iii). Bar = 100 nm.

shell domain of the opposing subunit such that α8 (at the C-terminus of one CP) is positioned next to the N-terminal α1 of the same monomer (Fig. 3a). This is a common feature of all Partitivirus structures solved to date, with PsV-F and PsV-S both having a C-terminal helix that domain swaps and sits next to the N-terminus (Fig. 3b, c).

**The disordered domain of deltapartitivirus CPs is hypervariable and not required for assembly.** Sequence comparison within the deltapartitiviruses shows that while the buried residues of the P domain are conserved, the disordered extension to the protrusions is highly variable (Fig. 4a). Variability extends to the surface-exposed loops of the P domain. We have compiled an additional 20 coat protein sequences likely belonging to the deltapartitiviruses and related to PCV-1 CP (Supplementary Fig. 5). Some deltapartitiviruses, such as Fig cryptic virus, which is assigned to the genus, as well as numerous others, possess a short form putative CP that share very little sequence homology with long form CPs (Supplementary Fig. 5). There is no evidence that these short form CPs assemble into capsids[6,29] and they were excluded from further analysis. Alignment of the amino acid sequences of all putative long form deltapartitivirus CPs shows a pattern of sequence conservation consistent with that from within the recognised genus members (Supplementary Fig. 6 and Fig. 4). The non-conserved region corresponding to the disordered extension to the protrusion shows very little similarity in sequence or in length (Supplementary Fig. 6). This hypervariable region is also predicted to be disordered for all the putative long form deltapartitiviruses identified (Supplementary Fig. 7). Together this indicates that the region of missing electron density for PCV-1 is likely to arise from intrinsic disorder of this region rather than flexibility of an ordered domain or motif. Moreover, the presence of an internal disordered region appears to be specific to the deltapartitiviruses as no similar disorder is predicted for partitiviruses of any other genus.

The S domain is relatively conserved between deltapartitiviruses with some variability at exposed loops (Fig. 4 and

Supplementary Fig. 6). It is tempting to speculate that the shell domain provides an evolutionarily stable framework around which deltapartitiviruses have diversified, much like in some fungal dsRNA viruses[12]. The predicted disordered region varies from zero to 119 aa among the deltapartitiviruses (Supplementary Fig. 7) suggesting that it is not essential for assembly. To test this, we substituted 48 aa of the region missing in the PCV-1 electron density with short glycine-based peptide linkers designed to provide flexibility to bridge the missing density as well as resistance to proteolysis in plants[30]. Transient expression in plants of two variants of the linker resulted in assembly of PCV-1 VLPs (Fig. 4c). The assembly of VLPs from these deletion mutants demonstrates the expendability of the disordered part of the P domain in assembly. This does not appear to be the case for any part of the P domain from the fungal partitiviruses investigated thus far, as the tip of the protrusion comprises part of the dimerisation interface (Fig. 3)[7,27].

Intrinsically disordered proteins and protein domains contribute a wide array of functions via interactions with other proteins, nucleic acids and other cellular components[31]. Specific to viral CPs, intrinsically disordered regions can be involved in capsid assembly and stability, regulating genome encapsidation, and capsid maturation including structural transitions and autoproteolytic cleavage[32]. A number of dsRNA viruses have evolved catalytic coat proteins[12,14] and intrinsic disorder is known to play a role in enzyme efficiency and evolution[33]. Intrinsically disordered domains are also central to many signalling and regulatory pathways including regulation of transcription and translation[34]. In particular, extended regions of disorder are characteristic of transcription factors[35]. This may provide a provocative clue as to how partitiviruses are able to influence the metabolism of their hosts. All but one long form deltapartitivirus coat proteins possess a conserved lysine residue alongside 1 to 5 additional basic residues within the disordered region. Polybasic motifs are characteristic of nuclear localisation signals[36] and it will be fascinating to investigate the subcellular localisation and possible interactions of PCV-1 CP using the transient expression tools we have developed.

The maintenance of the hypervariable region in the deltapartitivirus genus supports a hypothesis that it enables the acquisition of additional functions. Furthermore, redundancy of the disordered region of the P domain with respect to capsid assembly, and the intrinsic disorder of this region, provide a structural mechanism. It has been proposed that intrinsically disordered regions of proteins can provide a structural buffer against unfavourable mutations[37], enabling rapid evolution and protein diversification in viral proteins[38,39]. Deltapartitiviruses such as PCV-1 appear to play a complex role in the ecology of their hosts, changing secondary metabolism[15] of the host as part of their intimate mutualistic relationship. It is tempting to speculate that the presence of the disordered and hypervariable region in deltapartitiviruses represents the acquisition of CP functions in an adaptation to plant hosts.

## Conclusions

It has recently emerged that persistent viruses of plants may be beneficial to their infected hosts. Among them, the partitiviruses are some of the most common viruses in wild plants and as they have been described in a wide diversity of food crops, it is possible that partitiviruses like PCV-1 may have been positively selected during the domestication of horticultural varieties. The capsid structure presented here shows that while the S domain conserved among the plant-specific deltapartitiviruses shares some structural features with fungal partitiviruses, other elements of its capsid architecture are strikingly different. In particular, the P domain emerges from the capsid surface at the homodimer interface and has a disordered apex, which is hypervariable and does not play an essential role in assembly. Given the apparent lack of horizontal transmission of partitiviruses, the origins of plant-specific partitiviruses are not clear. Although a close relationship between plant and fungal partitiviruses has been established[40], it appears that deltapartitiviruses may be more closely related to partitiviruses from herbivorous arthropods than to fungal partitiviruses[41]. In addition to infecting plants, fungi and protozoa, partitivirus-like sequences have now been found to be common in wild populations of arthropods[42] where they are also vertically transmitted[41,43]. Our results identifying a hypervariable and intrinsically disordered domain among deltapartitiviruses may suggest a possible evolutionary mechanism that enables them to maintain persistent infections in diverse host plants. This remains to be experimentally tested and investigating the molecular determinants of the mutualistic relationships between partitiviruses and their hosts will be a particularly fascinating future area of study.

## Methods

**Expression and purification of PCV-1 VLPs**. The wild-type PCV-1 coding sequence (YP_009466860.1) and disordered region replacement variants were codon optimised for expression in *Nicotiana benthamiana* using the GeneArt web portal (www.lifetechnologies.com) and inserted into the pEAQ-HT plasmid (GQ497234)[17] by Gibson assembly. pEAQ-HT was linearised using *Age*I and *Stu*I restriction enzymes and recombinant plasmid constructs were verified with Sanger sequencing.

Binary expression vectors were maintained in *Agrobacterium tumefaciens* strain LBA4404 transformed by electroporation and propagated at 28 °C in the presence of 50 µg/L of kanamycin, streptomycin and rifampicin. LBA4404 cultures were resuspended in infiltration buffer (10 mM MES (pH 5.6) with 10 mM MgCl₂ and 100 µM acetosyringone) and incubated for 2 h at ambient temperature. Leaves of 6–7-week-old *Nicotiana benthamiana* plants were pressure infiltrated using 1 mL needleless syringe and incubated for 4–6 days. Approximately 18 g of infiltrated tissue was extracted in 50 mL extraction buffer (50 M MOPS (pH 7.0) with 140 M NaCl 0.1% (w/v) N-lauroylsarcosine sodium salt, 1 mM dithiothreitol) with complete EDTA-free protease inhibitor cocktail (www.sigmaaldrich.com). The cell lysate was filtered using miracloth to remove debris and centrifuged at 18,000 g for 15 min at 4 °C. Discontinuous gradients of 20, 30, 35, 40 and 50% iodixanol in PCV-1 buffer (20 mM MOPS, pH 7.0 with 140 mM NaCl) were used to sediment PCV-1 VLPs at 125,000 g for 3 h at 15 °C. Samples containing PCV1 were subjected to buffer exchange using PD MidiTrap G-25 desalting columns (www.cytivalifesciences.com) with PCV1 desalting buffer (20 mM MOPS, pH 7.0 with 140 mM NaCl and 5% (v/v) glycerol) and concentrated using Amicon Ultra-0.5 mL 100k Centrifugal Filters (www.merckmillipore.com).

**Negative stain electron microscopy**. Carbon-coated copper negative stain grids were first glow-discharged for 30 s (easiGlow, www.tedpella.com). Three ul of sample was applied and allowed to incubate for 30 s before blotting away excess liquid. Grids were washed twice in water before applying 3 ul of 2% w/v uranyl acetate. Stain was left for 30 s and blotted away. Grids were allowed to air dry and analysed using an FEI Tecnai F20 equipped with an FEI CMOS camera (Astbury Biostructure Laboratory, University of Leeds)

**Cryo-EM imaging**. Cryo-EM grids were first glow discharged for 30 s (easiGlow, www.tedpella.com). Samples were prepared in a chamber held at 95–100% relative humidity and 4 °C. 3 ul of purified PCV-1 VLPs were applied to 400 mesh lacey grids with an ultra-thin carbon support. The sample was left to incubate on the grid for 1 min and blotted using a FEI vitrobot mark IV (www.thermofisher.com). Grids were vitrified in liquid ethane cooled by liquid nitrogen. Data was collected on a ThermoFisher Titan Krios electron microscope (Astbury Biostructure Laboratory, University of Leeds) at 300 kV. Images were taken at 75,000x magnification, with a total electron dose of 66.1 e-/Å². Exposures were recorded using the EPU automated acquisition software on a Falcon III detector. Movies had a total exposure time of 1.5 s split across 59 fractions. Detailed parameters for data collection are shown in Table 1. Data collection protocol was derived from that described in Thompson et al.[44].

**Image processing**. Image processing statistics are shown in Table 1. Image processing was carried out with RELION 3.0[45]. Movie motion correction was carried out with MOTIONCOR2 and the contrast transfer function of each drift averaged micrograph determined using gCTF[46,47]. Approximately 1000 particles were picked manually and used to derive 2d class averages, which were then used as references for automated particle picking. Approximately 260,000 particles were picked. Particles were extracted in a 512 × 512 pixel box. Iterative 2D classification, and subset selection of the best 2d class averages reduced the particle stack to approximately 105,000 particles. An initial model was generated from a subset of these particles and used as a reference for 3D Refinement. Following initial refinement, per particle CTF refinement and Bayesian polishing were carried out. The final 3D reconstruction yielded a 2.9 Å resolution map.

**Table 1 Cryo-EM data collection, refinement and validation statistics.**

| | PCV-1 VLP |
|---|---|
| **Data collection and processing** | |
| Sample applications to grid | 1 |
| Magnification | 75, 000 x |
| Voltage (kV) | 300 |
| Electron exposure (e–/Å²) | 66.11 |
| Defocus range of micrographs (µm) | −0.5 to −2.0 |
| Pixel size (Å) | 1.065 |
| Symmetry imposed | I |
| Initial particle images (no.) | 260, 000 |
| Final particle images (no.) | 105, 000 |
| Map resolution (Å) | 2.9 |
| FSC threshold | 0.143 |
| Number of frames | 59 |
| **Refinement** | |
| Map sharpening *B* factor (Å²) | −127.7 |
| Model composition | |
| Protein residues | 668 |
| Nucleic acids | 0 |
| R.m.s. deviations | |
| Bond lengths (Å) | 0.006 |
| Bond angles (°) | 0.664 |
| Validation | |
| Clashscore | 8.29 |
| Poor rotamers (%) | 8.69 |
| Ramachandran plot | |
| Favoured (%) | 91.36 |
| Allowed (%) | 8.18 |
| Disallowed (%) | 0.45 |

**Model building**. The model for PCV-1 was built de novo into the final sharpened map using COOT[48]. Refinement was carried out with Phenix real space refine[49]. A single asymmetric unit of PCV-1, comprising a homodimer of PCV-1 was refined in the presence of all surrounding protomer chains to ensure inter-chain interactions were satisfied.

**Sequence analysis**. Additional long form deltapartitivirus sequences were identified using position-specific iterated basic local alignment search tool (PSI-BLAST[50]) using PCV-1 CP as the query. The search was limited to viral sequences and convergence was reached at 2 iterations. Clustal Omega[51] was performed using Geneious Prime 2019.2.3. Predictions for regions of protein disorder were made using the webserver for NetSurfP-2.0[52].

**Reporting summary**. Further information on research design is available in the Nature Research Reporting Summary linked to this article.

## Data availability

Coordinates are deposited in the Protein Data Bank under accession code 7NCR. Cryo-EM reconstruction is deposited in the EM Data Bank under accession code EMD-12270. All reagents and relevant data will be available from the authors upon reasonable request.

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

## Acknowledgements

The authors acknowledge the facilities, and the scientific and technical assistance, of the Microscopy Australia Facility at the Centre for Microscopy and Microanalysis (CMM), The University of Queensland. Aspects of this research have been facilitated by access to the Australian Proteome Analysis Facility supported under the Australian Government's National Collaborative Research Infrastructure Strategy (NCRIS). F.S. acknowledges the support of the CSIRO Synthetic Biology Future Science Platform. This work was supported by the UK Biotechnological and Biological Sciences Research Council BB/R00160X/1. We thank the Astbury Biostructure Laboratory (ABSL), for assisting with cryo-EM data collection. All electron microscopy was performed at ABSL which was funded by the University of Leeds and the Wellcome Trust (108466/Z/15/Z).

## Author contributions

M.J.B., A.K., L.E. and F.S. carried out the experiments and analysed data. F.S., M.J.B. and N.A.R. conceived and directed the project. All authors contributed to the writing and editing of the manuscript.

## Competing interests

F.S. declares that he is a named inventor on granted patent WO 29087391 A1 which describes the HyperTrans (HT) expression system and associated pEAQ vectors (pEAQ-HT) used in this manuscript. All other authors declare no competing interests.
