## [Transparent Peer Review File · Communications Biology]

Reviewers' comments:

Reviewer #1 (Remarks to the Author):

The manuscript by Byrne and others reports the nucleocapsid structure of Pepper cryptic virus 1 (PCV-1) determined using virus-like particles obtained upon transient expression of viral capsid protein (CP) in *N. benthamiana* plants. This is the first structure reported for plant partitiviruses, it describes interesting novel features of PCV-1 CP, and is definitely worth publishing. To my mind, *Communications Biology* is quite appropriate journal for publication of this paper. However, I have some concerns and suggestions, which should be addressed before the paper can be accepted for publication.

1. Title.

First, the authors claim that the PCV-1 CP has "novel coat protein architecture". This is not correct. The PCV-1 CP, as the authors clearly show in their paper, is structurally closely related to CPs of other partitiviruses, both at the primary structure level (Fig. 4A) and at the level of conservation of key alpha-helices in the S domain (Fig. 3). In the PCV-1 CP, novel is the structure of P domain only, therefore the title should be changed accordingly.

Second, I would avoid saying "hypervariable region". The title refers to PCV-1 CP only, and this wording may be interpreted as that this region of PCV-1 CP is extremely variable. On the other hand, it would be useful to say in the title that the CP contains intrinsically disordered region, rather than a region which is not conserved among partitiviruses.

Therefore, I would suggest the following corrected title: Structure of a plant-specific partitivirus capsid reveals a novel architecture of coat protein protruding domain comprising an intrinsically disordered region.

2. When discussing intrinsically disordered domain in viral proteins, papers by V. Uversky should be cited, as he has published more than 80 papers on viral disordered domains, including those in virus capsid proteins. The reported findings should be discussed in the context of current knowledge in the field.

3. The last phrase before Conclusions is fully speculative and confusing. What are links between (i) the fact that this protein region is intrinsically disordered, (ii) observation that it contains charged residues, and (iii) protein subcellular localization? I would delete this phrase and, instead, discuss the role intrinsically disordered regions in other virus CPs in comparison to

4. The Conclusion section contains a number of unjustified/incorrect/inadequate statements. These phrases should be re-written or deleted.

1) "The restriction of the disordered region to the plant-limited deltapartitiviruses also suggests that it represents an adaptation to plants that is not compatible with fungal hosts."

The presence of this region in plant-specific but not other partitiviruses does not mean that this region represent an adaptation to plant hosts, and this may be just a correlation. It would be also very interesting to know why the authors think that a specific adaptation to plant cells is necessary for viruses, which do not require host cell-encoded protein cofactors and, of all possible cell resources, use only nucleotide triphosphates and the translation machinery. If the authors see definite reasons for the "adaptation to plant hosts", these reasons should be explained.

2) "Given the apparent lack of horizontal transmission of partitiviruses, it is not clear how deltapartitiviruses found their plant hosts"

Such student-style wording is not suitable for scientific literature. Besides, possible evolutionary origin of plant partitiviruses is discussed in papers by M. Roossinck, including those cited in this paper.

3) "Our results point to a possible evolutionary mechanism to increase the density of encoded functions that enables them to maintain long term infections in diverse host plants. The relatively simple genome and proteome of partitiviruses is in stark contrast to these complex roles in the ecology and evolution of their hosts."

Several strange statements in a row. Acquisition of additional protein domains does not mean an increase in 'density of functions' (although this term itself looks very strange). It is not clear why the reported results point to a possible mechanism of acquisition of additional sequences and what mechanism authors mean. Not clear why the authors see a link between acquisition of additional domain and "long term" (persistent?) infections, as many partitiviruses lacking such domains infect fungi and other organisms in a persistent manner. Not clear why "diverse host plants" are mentioned. The word "proteome" is not appropriate to describe two proteins encoded by partitivirus genome. And finally it is absolutely incorrect to say that plant partitiviruses has "complex roles in the ecology and evolution of their hosts", as such roles were not demonstrated. To my mind, the Conclusion section should be re-written.

Minor points:

Introduction, 3rd paragraph:

"the partitiviruses must maintain a functional capsid to prevent the recognition of the dsRNA genome by the host" - not fully correct: not only to prevent dsRNA recognition, but also for genome replication, which takes place inside the capsid containing viral RNA replicase.

Fig.3 legend

cooured should be coloured

Subtitle: The disordered domain is hypervariable and not required for assembly.

At least "in partitiviruses" should be added after hypervariable.

Also, what is the difference between variable and hypervariable sequences? What is definition of hypervariable sequences? I would simply say "non-conserved".

Reviewer #2 (Remarks to the Author):

The manuscript "Structure of a plant-specific partitivirus capsid reveals a novel coat protein architecture with a hypervariable protrusion" by Byrne et al. describes the 2.9 Å resolution structure of the pepper cryptic virus 1 (PCV-1) capsid protein from empty VLPs. This viral capsid is formed by 60 capsid protein (CP) homodimers with a T=1 symmetry. The CP structure is compared with other members of the family Partitivirus that infect fungi and with related dsRNA viruses such as picobirnaviruses. Whereas the shell domain of PCV-1 CP is relatively conserved among partitiviruses, the protrusion domain has a hypervariable, intrinsically disordered region at the interface of the two dimers that contrasts with the arch-like arrangements found in partitiviruses. This hypervariable region, which is not required for assembly, might be related with the mutualistic relation between the host plant and the virus, providing a permissive site for insertions that has favored this symbiotic relationship. I recommend publication of this manuscript after introduction of a few changes to overcome some problems in the present version.

Comments:

1. Although optional, the virus name is omitted in the title, but at least it should be mentioned once in the abstract. Also, it is referred to as PCV1 or PCV-1 (please, use the correct abbreviation).
2. The hypervariable region (segment 329-378) has no density for tracing. I wonder if the authors have made any SSE prediction for this segment and if they looked for any match/similarity with known motifs/domains. Any speculations about the reasons for the missed 1-29 segment?
3. There exist a color correspondence problem between for SSE in Fig. S4 (panels A and B) and the respective descriptions in the main text. The $\alpha 7$ and $\alpha 8$ helices do not match with their associated

colors. Furthermore, Fig. 3 does not follow the same color code as Fig. S4. To avoid confusion, these major helices should be clearly labelled in the panels.

4. An explanation/context should be given for the two linker sequences that were chosen for the protrusion deletions. In addition to these deletion mutants, it seems promising to use this flexible region to insert longer peptides/domains with testable activities. Fig. 4C provides a piece of evidence of the dispensability of the hypervariable region for capsid assembly. However, the EM images show a limited field of assembled wt and mutant particles. Any quantitative evaluation about the assembly process would be valuable. In any case, these images should be substituted for higher quality images; panels i and iii seems parts of an immunogold labeling experiment, probably due the uranyl acetate precipitations around the VLPs, and panel ii seems a clear example of positive stained VLPs.

5. Revise references section as many cites are incomplete, with/without abbreviations, or contain mistakes (Ref 35 is unintelligible).

Minor comments:

In the methods section (negative stain electron microscopy), microliters are noted as "u" instead of "μ".

We are very glad to hear the overall positive assessment about the importance and quality of our work. Below we have provided a detailed point-by-point response to the comments and issues raised.

Reviewer #1 (Remarks to the Author):

The manuscript by Byrne and others reports the nucleocapsid structure of Pepper cryptic virus 1 (PCV-1) determined using virus-like particles obtained upon transient expression of viral capsid protein (CP) in *N.benthamiana* plants. This is the first structure reported for plant partitiviruses, it describes interesting novel features of PCV-1 CP, and is definitely worth publishing. To my mind, Communications Biology is quite appropriate journal for publication of this paper. However, I have some concerns and suggestions, which should be addressed before the paper can be accepted for publication.

1. Title.

First, the authors claim that the PCV-1 CP has "novel coat protein architecture". This is not correct. The PCV-1 CP, as the authors clearly show in their paper, is structurally closely related to CPs of other partitiviruses, both at the primary structure level (Fig. 4A) and at the level of conservation of key alpha-helices in the S domain (Fig. 3). In the PCV-1 CP, novel is the structure of P domain only, therefore the title should be changed accordingly.

Second, I would avoid saying "hypervariable region". The title refers to PCV-1 CP only, and this wording may be interpreted as that this region of PCV-1 CP is extremely variable. On the other hand, it would be useful to say in the title that the CP contains intrinsically disordered region, rather than a region which is not conserved among partitiviruses.

Therefore, I would suggest the following corrected title: Structure of a plant-specific partivirus capsid reveals a novel architecture of coat protein protruding domain comprising an intrinsically disordered region.

We appreciate the attention to detail in this comment. While we believe our identification of a novel coat protein architecture is correct, we agree with the reviewers that it is specifically the *domain* architecture. The emergence of the P domain at the dimerization interface and the extent of its contribution to the buried surface area at this interface are unique. Figure 4 is a comparison of deltapartitiviruses only and that is why a moderate level of similarity is found. It does not show primary structure similarity with partitiviruses of other genera. Thus, despite conserved features of the S domain, PCV-1 clearly comprises a novel tertiary structure and domain architecture of the CP, distinct from other partivirus genera and from other dsRNA viruses. We also agree with the possible misinterpretation raised by the reviewer and that it would be useful to highlight the intrinsically disordered region in the title. Therefore, we have changed the title to "The Structure of a Plant-Specific Partivirus Capsid Reveals a Novel Coat Protein Domain Architecture with an Intrinsically Disordered Protrusion"

2. When discussing intrinsically disordered domain in viral proteins, papers by V.Uversky should be cited, as he has published more than 80 papers on viral disordered domains, including those in virus capsid proteins. The reported findings should be discussed in the context of current knowledge in the field.

In light of the reviewer's other comments regarding the conclusion, which we took the opportunity to revisit, this is a very good suggestion. Beyond the general features or disordered proteins already

discussed, Uvesrsky's ideas are very useful in clarifying our proposed function of the deltapartivirus disordered region as an evolutionary mechanism to adapt to plant hosts.

In addition, we have added the following context for disordered proteins or protein regions of viral coat proteins: "Specific to viral CPs, intrinsically disordered regions can be involved in capsid assembly and stability, regulating genome encapsidation, and capsid maturation including structural transitions and autoproteolytic cleavage."

3. The last phrase before Conclusions is fully speculative and confusing. What are links between (i) the fact that this protein region is intrinsically disordered, (ii) observation that it contains charged residues, and (iii) protein subcellular localization? I would delete this phrase and, instead, discuss the role intrinsically disordered regions in other virus CPs in comparison to

The reviewer is right in that this is a confusingly written. We make no apologies for the fact that this passage is speculative, but we have added an explanation of the link between commonly found disorder in transcription factors, polybasic motifs and subcellular localisation by adding the phrase "Polybasic motifs are characteristic of nuclear localisation signals and nucleic acid binding domains..." to preface the interest in investigating subcellular localisation.

4. The Conclusion section contains a number of unjustified/incorrect/inadequate statements. These phrases should be re-written or deleted.

On consideration of the reviewers comments we have taken the opportunity to strengthen the conclusions section with a number of changes, including moving the section that discusses a possible evolutionary mechanism for the acquisition of additional functions to the end of the discussion. The final paragraph of the discussion is now:

"The maintenance of the hypervariable region in the deltapartivirus genus supports a hypothesis that it enables the acquisition of additional functions. Furthermore, redundancy of the disordered region of the P domain with respect to capsid assembly, and the intrinsic disorder of this region, provide a structural mechanism. It has been proposed that intrinsically disordered regions of proteins can provide a structural buffer against unfavourable mutations, enabling rapid evolution and protein diversification in viral proteins. Deltapartiviruses such as PCV-1 appear to play a complex role in the ecology of their hosts, changing secondary metabolism of the host as part of their intimate mutualistic relationship. It is tempting to speculate that the presence of the disordered and hypervariable region in deltapartiviruses represents the acquisition of CP functions in an adaptation to their plant hosts."

1) "The restriction of the disordered region to the plant-limited deltapartiviruses also suggests that it represents an adaptation to plants that is not compatible with fungal hosts."

The presence of this region in plant-specific but not other partitiviruses does not mean that this region represent an adaptation to plant hosts, and this may be just a correlation. It would be also very interesting to know why the authors think that a specific adaptation to plant cells is necessary for viruses, which do not require host cell-encoded protein cofactors and, of all possible cell resources, use only nucleotide triphosphates and the translation machinery. If the authors see definite reasons for the "adaptation to plant hosts", these reasons should be explained.

This particular sentence was not clear and was meant to simply refer to the fact that deltapartiviruses are plant specific. It was irrelevant and has now been removed.

Although it is not known whether partitiviruses require host cell protein co-factors, it has certainly been shown that PCV-1 has an impact on host physiology. We think that it is now clearer, in what is now the final paragraph of our discussion, that we are speculating that the disordered region may play a role in modifying plant cell physiology and that this may contribute to the maintenance of persistent infection.

2) “Given the apparent lack of horizontal transmission of partitiviruses, it is not clear how deltapartitiviruses found their plant hosts”

Such student-style wording is not suitable for scientific literature. Besides, possible evolutionary origin of plant partitiviruses is discussed in papers by M. Roossinck, including those cited in this paper.

While this comment is not the sort of constructive comment one hopes for, we agree that the phrase is rather ambiguous. We have changed the text to read “Given the apparent lack of horizontal transmission of partitiviruses, the origins of plant-specific partitiviruses are not clear.”

As the reviewer states, the papers we cite point to a *possible* fungal origin for plant partitiviruses and we have left the relevant Roossinck citation to indicate the close relationship with fungal partitiviruses. It also appears that deltapartitiviruses may be more closely related to partitiviruses from arthropods and we have added this to the discussion as follows: “Although a close relationship between plant and fungal partitiviruses has been established, it appears that deltapartitiviruses may be more closely related to partitiviruses from herbivorous arthropods than to fungal partitiviruses.”

3) “Our results point to a possible evolutionary mechanism to increase the density of encoded functions that enables them to maintain long term infections in diverse host plants. The relatively simple genome and proteome of partitiviruses is in stark contrast to these complex roles in the ecology and evolution of their hosts.”

Several strange statements in a row. Acquisition of additional protein domains does not mean an increase in ‘density of functions’ (although this term itself looks very strange).

This is true and we have removed this somewhat erroneous reference to the concept of gene density.

It is not clear why the reported results point to a possible mechanism of acquisition of additional sequences and what mechanism authors mean. Not clear why the authors see a link between acquisition of additional domain and “long term” (persistent?) infections, as many partitiviruses lacking such domains infect fungi and other organisms in a persistent manner.

We hope that we have now made clear in the final part of the discussion that we are speculating that there is a link between the intrinsically disordered domain and deltapartitivirus-host interactions.

Here the use “long term” infection was indeed a clumsy reference to persistent infections. As we are only talking about persistent viruses as the reviewer quite rightly points out, this ambiguity has been clarified to state that we are referring specifically to the mutualistic infection of plants.

Not clear why “diverse host plants” are mentioned. The word “proteome” is not appropriate to describe two proteins encoded by partitivirus genome. And finally it is absolutely incorrect to say that plant partitiruses has “complex roles in the ecology and evolution of their hosts”, as such roles were not demonstrated.

The 24 long-form deltapartitivirus CPs we identify are from 23 different host species. It is striking that there are no recorded incidences of a deltapartitivirus infecting multiple plant species, indicating a high degree of specialisation.

The sentence where proteome is used and there is a reference to the complex roles of deltapartitiruses in the ecology of their hosts is no longer required in the re-written conclusions section and has been deleted.

To my mind, the Conclusion section should be re-written.

As stated above, we believe it has now been strengthened and clarified.

Minor points:

Introduction, 3rd paragraph:

“the partitiruses must maintain a functional capsid to prevent the recognition of the dsRNA genome by the host” - not fully correct: not only to prevent dsRNA recognition, but also for genome replication, which takes place inside the capsid containing viral RNA replicase.

We agree that this is not complete and have changed to the following: “...the partitiruses must maintain a functional and intact capsid to prevent host recognition of the dsRNA genome and co-localise the RdRp in a protective compartment for genome replication”

Fig.3 legend

coored should be coloured

Thanks to the reviewer for spotting this, it has been changed.

Subtitle: The disordered domain is hypervariable and not required for assembly.

At least “in partitiruses” should be added after hypervariable.

Also, what is the difference between variable and hypervariable sequences? What is definition of hypervariable sequences? I would simply say “non-conserved”.

We agree that this subheading should specify deltapartitiruses and have changed to the following: “The disordered domain of deltapartitivirus CPs is hypervariable and not required for assembly”

In order to more precisely define hypervariable we have modified the text to explicitly state that it is a region of very little conservation in sequence and in length: “The non-conserved region corresponding to the disordered extension to the protrusion shows very little similarity in sequence or in length (Figure S6). This hypervariable region is also predicted to be disordered for all the putative long form deltapartitiruses identified (Figure S7).”

Reviewer #2 (Remarks to the Author):

The manuscript "Structure of a plant-specific partitivirus capsid reveals a novel coat protein architecture with a hypervariable protrusion" by Byrne et al. describes the 2.9 Å resolution structure of the pepper cryptic virus 1 (PCV-1) capsid protein from empty VLPs. This viral capsid is formed by 60 capsid protein (CP) homodimers with a T=1 symmetry. The CP structure is compared with other members of the family Partitivirus that infect fungi and with related dsRNA viruses such as picobirnaviruses. Whereas the shell domain of PCV-1 CP is relatively conserved among partitiviruses, the protrusion domain has a hypervariable, intrinsically disordered region at the interface of the two dimers that contrasts with the arch-like arrangements found in partitiviruses. This hypervariable region, which is not required for assembly, might be related with the mutualistic relation between the host plant and the virus, providing a permissive site for insertions that has favored this symbiotic relationship. I recommend publication of this manuscript after introduction of a few changes to overcome some problems in the present version.

Comments:

1. Although optional, the virus name is omitted in the title, but at least it should be mentioned once in the abstract. Also, it is referred to as PCV1 or PCV-1 (please, use the correct abbreviation).

We agree with this comment and have put Pepper cryptic virus 1 in the abstract. Thanks to the reviewer for spotting the inconsistencies, all instances of PCV1 have been corrected to PCV-1.

2. The hypervariable region (segment 329-378) has no density for tracing. I wonder if the authors have made any SSE prediction for this segment and if they looked for any match/similarity with known motifs/domains. Any speculations about the reasons for the missed 1-29 segment?

Bioinformatic analysis of the secondary structure here also predicts it to be disordered as shown in supplementary figure S7, and so our interpretation for the fact that it is missing, is that it is intrinsically disordered. Furthermore, database searches find no similar motifs or domains for this region of PCV-1, or any of the putative disordered regions of the other long-form deltapartitivirus CPs, among known proteins.

We have added a short description for the structure at the N-terminus, which we neglected to mention: "Likewise, there is no density for the N-terminus (residues 1-28), indicating that this region is also highly flexible."

3. There exist a color correspondence problem between for SSE in Fig. S4 (panels A and B) and the respective descriptions in the main text. The $\alpha 7$ and $\alpha 8$ helices do not match with their associated colors. Furthermore, Fig. 3 does not follow the same color code as Fig. S4. To avoid confusion, these major helices should be clearly labelled in the panels.

We thank the reviewer for spotting this and have replaced figure S4 with one that has better colouring and labels for the secondary structure elements in the top panel. Low resolution version shown here:

A

B

4. An explanation/context should be given for the two linker sequences that were chosen for the protrusion deletions. In addition to these deletion mutants, it seems promising to use this flexible region to insert longer peptides/domains with testable activities. Fig. 4C provides a piece of evidence of the dispensability of the hypervariable region for capsid assembly. However, the EM images show a limited field of assembled wt and mutant particles. Any quantitative evaluation about the assembly process would be valuable. In any case, these images should be substituted for higher quality images; panels i and iii seems parts of an immunogold labeling experiment, probably due the uranyl acetate precipitations around the VLPs, and panel ii seems a clear example of positive stained VLPs.

The linkers are based on a synthetic design, (Gly2-4Ser)_n, that is generally known to be flexible. We agree with the reviewer's assertion that more information would be useful. We have now indicated that these are synthetic designs intended to have the flexibility and to be clear that they do not come from any other source and add reference in the manuscript for further information.

"...we substituted 48 aa of the region missing in the PCV-1 electron density with short glycine-based peptide linkers designed to provide flexibility to bridge the missing density as well as resistance to proteolysis in plants."

The observation that the shorter of the two linkers resulted in reduced yields indicates that a certain level of flexibility is required for assembly and we have also added a comment to this effect.

As the reviewer states, the result in the section shows that assembly of the mutants is possible and, therefore, that the disordered region is not required for assembly. A quantitative analysis of this, which would require more particles imaged alongside determination of the process yields on a biomass basis, would indeed be useful for a further study on the use of PCV-1 particles to present more interesting peptides/domains. However, that is outside the scope of this paper.

We agree that there are some staining artefacts in these images likely from uranyl acetate precipitates, which is not unusual, but in each case the protein of the capsid is shown as negatively stained. Our view is that this does not affect the results presented in figure 4, nor the conclusions of the study.

5. Revise references section as many cites are incomplete, with/without abbreviations, or contain mistakes (Ref 35 is unintelligible).

Thank you, the formatting issues in some of the references have been fixed.

Minor comments:

In the methods section (negative stain electron microscopy), microliters are noted as “u” instead of “μ”.

Thanks to the reviewer for spotting this, it has been changed.

REVIEWERS' COMMENTS:

Reviewer #1 (Remarks to the Author):

The paper has been considerably improved, and the revised version can be accepted for publication.

Reviewer #2 (Remarks to the Author):

Authors have addressed most of raised concerns of my first revision, except point 4 relative to the negative stained images shown in fig. 4c